



# Wind Vane Correction during Yaw Misalignment for Horizontal Axis Wind Turbines

Andreas Rott[1], Leo Höning[1,2], Paul Hulsmann[1], Laura J. Lukassen[1], Christof Moldenhauer[3], and Martin Kühn[1]

[1]ForWind, Institute of Physics, Carl von Ossietzky University Oldenburg, Küpkersweg 70, 26129 Oldenburg, Germany
[2]Fraunhofer Institute for Wind Energy Systems, Küpkersweg 70, 26129 Oldenburg, Germany
[3]Ocean Breeze Energy GmbH & Co. KG, Am Freihafen 1, 26725 Emden, Germany

**Correspondence:** Andreas Rott (andreas.rott@uol.de)

**Abstract.** This paper investigates the accuracy of wind direction measurements for horizontal axis wind turbines and its impact on yaw control. The yaw controller is crucial for aligning the rotor with the wind direction and optimizing energy extraction. Wind direction is conventionally measured by one or two wind vanes located on the nacelle, but the proximity of the rotor can interfere with these measurements. The authors show that the conventional corrections, including low-pass filters and calibrated

offset correction, are not adequate to account for a systematic overestimation of the wind direction deviation caused by the rotor misalignment. This measurement error can lead to an overcorrection of the yaw controller, and thus, to an oscillating yaw behaviour, even if the wind direction is relatively steady. The authors present a theoretical basis and methods for quantifying the wind vane measurement error and validate their findings using computational fluid dynamics simulations and operational data from two commercial wind turbines. Additionally, the authors propose a correction function that improves the wind vane

measurements and demonstrate its effectiveness in two free field experiments. Overall, the paper provides new insights into the accuracy of wind direction measurements and proposes solutions to improve the yaw control for horizontal axis wind turbines.

## 1   Introduction

Wind turbines are an increasingly important source of renewable energy, and their performance and efficiency are critical factors in their widespread adoption. One of the key parameters that affects horizontal axis wind turbine performance is the

alignment of the rotor with the incoming wind direction. The standard procedure for achieving such alignment involves the use of one or two wind vanes to detect deviations from the wind direction and adjust the yaw angle of the turbine through an active yaw manoeuvre accordingly. During commissioning the wind vanes are oriented along the rotor axis, followed by offset correction calibration to account for wake rotation over the nacelle, thereby achieving the most precise alignment of the wind turbine in the flow direction for a wind direction deviation of $0°$. The IEC 61400-12-3 standard for measurement

based site calibration (International Electrotechnical Commission, 2022) provides guidelines for the measurement, analysis, and reporting of site calibration in power performance testing for wind turbines. But due to the intricate flow field surrounding the turbine and associated wake effects this still poses a significant challenge. The question arises whether it is possible to reduce uncertainties and calibrate the wind vane in such a way that a better alignment of the wind turbine is achieved without





having to resort to additional or external measuring systems. In this way, on the one hand, a higher power yield could be
achieved and, on the other hand, the yaw activity could be reduced, which would protect the yaw motors and brakes and, thus,
increase their lifetime.

In (Kragh and Fleming, 2012) an amplification of the wind direction deviation behind the rotor of a test turbine was pointed
out, which is correlated to the rotor speed. A linear correction function depending on the rotor speed was presented, which was
used to improve the measured wind direction deviation of the test turbine. Mittelmeier and Kühn (2018) presented a three step
method based on SCADA data to detect changes in the wind turbine alignment during the operational lifetime and to improve
the alignment.

Additional temporary or permanent measurement devices, such as spinner anemometry by ultra sonic sensors (Pedersen
et al., 2014), nacelle-based lidars (Held and Mann, 2019) or monitoring of cyclic blade root bending moments (Bertelè et al.,
2017; Schreiber et al., 2020) have been proposed to improve the alignment.

A subject of extensive scientific research for several years has been the active wake deflection (Gebraad et al., 2016; Rott
et al., 2018; Bromm et al., 2018). Especially for this kind of control a well-calibrated wind vane is essential, as this method, in
particular, requires specific misalignments to be maintained.

Recently Simley et al. (2021) has reported that during experiments on a Senvion MM82 wind turbine, it was observed that
the wind vane overestimated the wind direction deviation compared to a nacelle-based lidar measurements. A linear wind-
speed-dependent transfer function was proposed to correct the wind vane. Simley et al. suggested that more complex functions
may be more appropriate for future yaw controllers and more research is needed in this area.

Nevertheless, many questions remain unanswered, such as occurrence and causes of deterministic errors of the wind vane
and how the wind vane could be calibrated or corrected to achieve better performance for regular operation or specific control
techniques such as active wake deflection.

In our study, we investigate the yaw behaviour of two commercial wind turbine types. We observe the wind vane signal
before and after a yaw manoeuvre and compare the obtained wind direction with a reference signal from a nearby measuring
mast. In addition, we conduct a multi-stage experiment with the wind turbine, in which we investigate different correction
functions for the wind vane. With this publication, we would like to find answers to the following questions:

1. Does a systematic wind vane error during yaw misalignment of common utility-scale wind turbines exist and how can it
   be characterised?

2. How can the wind vane be corrected based on operational data with and without external reference measurements?

3. What effects does a correction of the wind vane during yaw misalignment have on the performance of a wind turbine?

## 2 Methods

In this section, we first give a brief overview of the operation of a conventional wind turbine yaw controller (Section 2.1). Then
we outline our hypothesis about the causes leading to a measurement error of a wind vane behind the rotor of a wind turbine





and create a model for the error estimation (Section 2.2). Next, we detail the Computational Fluid Dynamics (CFD) simulation we performed to confirm our assumption (Section 2.3). In subsection (Section 2.4), we give a description of the free-field data we used and the experiments we performed with two commercial wind turbines.

## 2.1 General Yaw Control

To ensure that a wind turbine aligns with the wind direction, active yaw control is commonly applied. As exemplarily described in the Wind Energy Handbook (Burton et al., 2011), the yaw error measured by the wind vane on the nacelle is used to calculate a demand signal for the yaw actuator. To avoid the yaw control being influenced by small fluctuations, the measurement of the wind vane is averaged (e.g. moving averages with a window size of $30\,\mathrm{s}$, $60\,\mathrm{s}$ or even $180\,\mathrm{s}$ are commonly used). For the control, a dead-band controller is typically used, where a yaw manoeuvre is initiated if the yaw error exceeds a predefined threshold.

We refer to this threshold as the *yaw trigger*. With a standard yaw control, the magnitude of the yaw rotation corresponds to the determined yaw error. However, for special yaw strategies, this value may differ from the measured deviation. An example of this is the active wake deflection (Gebraad et al., 2016; Rott et al., 2018) mentioned above, in which a specific yaw misalignment is applied for certain wind direction sectors in a wind farm. Since the target value of the yaw manoeuvre is also adjusted in this study, we refer to this value as *yaw target*.

## 2.2 Wake Deflection


As a motivation for the investigation carried out in this study, we look at an example time series of the wind direction measurement by a wind vane on the nacelle of a commercial wind turbine. The wind vane measures the wind direction deviation from the nacelle's orientation, which we denote by $\varphi_{\mathrm{wt}}(t) \in [-180°, 180°)$, where $t \in \mathbb{R}$ represents the time. The measured wind direction in the global frame of reference $\omega_{\mathrm{wt}}(t) \in [0°, 360°)$ is the sum of the wind direction deviation $\varphi_{\mathrm{wt}}(t)$ and the

orientation of the nacelle (yaw angle) $\gamma_{\mathrm{wt}}(t) \in [0°, 360°)$ :

$$\omega_{\mathrm{wt}}(t) \equiv \varphi_{\mathrm{wt}}(t) + \gamma_{\mathrm{wt}}(t) \pmod{360°}. \tag{1}$$

For the sake of better readability, we will omit the modulo notation in the following and imply that angle specifications always lie in the value range $[-180°, 180°)$ for angular deviations and in the value range $[0°, 360°)$ for absolute angles. Figure 1 shows $\gamma_{\mathrm{wt}}(t), \omega_{\mathrm{wt}}(t)$, and the wind direction with a centred 60-s moving average:

$$\tilde{\omega}_{\mathrm{wt}}(t) = \frac{1}{60\,\mathrm{s}} \int\limits_{t-30\,\mathrm{s}}^{t+30\,\mathrm{s}} \omega_{\mathrm{wt}}(\tau)\mathrm{d}\tau. \tag{2}$$

It should be noted that we have used the arithmetic mean for the directional values here and also in the following, rather than the directional mean calculated over the vectorial components, even though this is not technically correct. We have chosen to use the arithmetic mean because this is more widely used for such calculations when dealing with Supervisory Control and Data Acquisition (SCADA) data, and because the difference is negligible for the relatively small directional values around $0°$,

as is the case here.



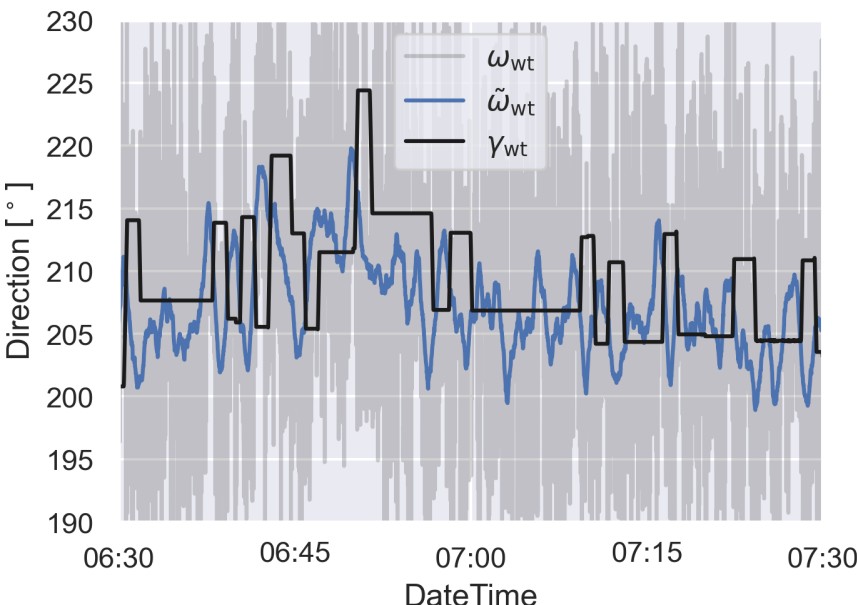

**Figure 1.** Example of a 60-min time series from a BARD 5.0 wind turbine of the nacelle orientation $\gamma_{\mathrm{wt}}$ (black) and the wind direction measured by the wind vane $\omega_{\mathrm{wt}}$ in 1-Hz resolution (grey). In addition, the wind direction was averaged with a centered 60-s moving window $\overline{\omega}_{\mathrm{wt}}$ (blue).

In Figure 1, it is noticeable that the BARD 5.0 wind turbine makes a relatively large number of yaw manoeuvres in the 60-min time series, even though the measured averaged wind direction $\tilde{\omega}_{\mathrm{wt}}(t)$ changes relatively little in the time period shown. In most cases, the directions of the yaw manoeuvre alternate, i.e. a clockwise rotation (yaw angle increases) is followed by a counterclockwise rotation of the nacelle (yaw angle decreases). Furthermore, although the wind direction looks like a highly

noisy stochastic process, i.e., it undergoes random and unpredictable changes, it seems that the moving average of the wind direction makes a turn in the opposite direction exactly in the situations where the wind turbine performs a yaw manoeuvre. In Layman's terms, it is almost as if the wind is trying to avoid the wind turbine. It should be noted, however, that the time series shown here has been deliberately chosen so that these features can be easily identified. Nevertheless, we could observe this behaviour very often when examining measurement data from different wind turbines, therefore, we presume that there is

causality between the measured wind direction changes and the turbine, which was the motivation for the present investigation.

Our hypothesis is that the same mechanism that causes the deflection of the intermediate to far wake during yaw misalignment (Jiménez et al., 2010; Bastankhah and Porté-Agel, 2016), i.e. the thrust component perpendicular to the inflow direction, is affecting also the wind direction measured a few metres behind the rotor plan on top of the nacelle. Such a deflection, though, would explain an overestimation of the wind direction deviation by a wind vane on the nacelle. Due to an overestimation of

the wind direction deviation, a yaw manoeuvre is triggered in the yaw algorithm earlier than intended and the orientation to





which the rotor adjusts overshoots the actual target. This increases the probability that the wind turbine now has an opposite yaw misalingment. If this resulting misalignment is again overestimated by the wind turbine, this can result in an alternating yaw pattern, where the wind turbine tries to follow the wind direction, but repeatetly overshoots.

In order to model the overestimation of the wind vane, we use a simple linear transfer function which approximates the relationship between the wind direction measured by the wind vane $\varphi_{\mathrm{wt}} \in [-180°, 180°)$ and an estimate (represented by the hat $\hat{\cdot}$) of the "true" or reference wind direction deviation $\varphi_{\mathrm{ref}} \in [-180°, 180°)$ :

$$\hat{\varphi}_{\mathrm{ref}} = c \cdot \varphi_{\mathrm{wt}} + b, \tag{3}$$

where $c, b \in \mathbb{R}$ are the parameters describing the slope and the offset respectively. The offset $b$ is attributed to mounting error of the wind vane and the rotation of the wake. It is usually determined during the calibration of the wind vane or by more elaborate analyses of the power performance (Mittelmeier and Kühn, 2018). Wind vane data therefore normally already includes a correction for this offset. Therefore, in the following we focus our analysis on the correction factor $c$ and set the offset factor $b = 0°$.

In the following sections, we will show how we estimate the correction factor $c$ using firstly the comparison between the turbine's wind vane and a met mast and secondly only the SCADA signal measurements at the turbine.

## 2.3 CFD Simulation Setup

For the purpose of verification, CFD simulations of the NREL 5MW reference turbine were performed (Jonkman et al., 2009) using the open-source CFD software OpenFOAM (OpenFOAM, 2021). The numerical grid was generated making use of the two in-house tools *bladeBlockMesher* and *windTurbineMesher* (Rahimi et al., 2016) and consist of 26.4 mio cells. The rectangular meshes contain the yawed rotor with a diameter of $D = 126 \, \mathrm{m}$ and a cylindrical, non-rotating nacelle geometry, neglecting the influence of the tower. The length of the nacelle was chosen to be $16 \, \mathrm{m}$ with a radius of $1.35 \, \mathrm{m}$. The rotor is located $5D$ from the inlet and $15D$ from the outlet, with a distance of $3.5D$ towards all sides. Extra mesh refinement in the vicinity of the nacelle and blade roots was made, ensuring that the flow at the probe locations is resolved reasonably well. The rotation of the rigid blades is accounted for using sliding mesh interfaces between the rotor and the farfield grids. Five different yaw angles were investigated, namely, $-20°, -10°, 0°, 10°$ and $20°$, with an inflow wind velocity of $11.4 \, \mathrm{m/s}$ and a constant rotational speed of $12.1 \, \mathrm{RPM}$.

The incompressible, transient flow was simulated using the hybrid Spalart-Allmaras delayed detached eddy simulation type (Spalart et al., 2006). To advance the solution in time, a second-order implicit backward method was used. Temporal discretization made use of a second-order accurate Gauss linear scheme.

On top of the nacelle, a total of 390 probes were placed at three different heights ($2.35 \, \mathrm{m}$, $2.93 \, \mathrm{m}$ and $3.50 \, \mathrm{m}$) above the axis of rotation, representing possible wind vane positions. Figure 2 provides a schematic view of the turbine and probe positions from both the top and the side, and Figure 3 displays a perspective view from the simulation. In both figures, the axis origin is shifted from the center of the rotor in front of the turbine to make it visible





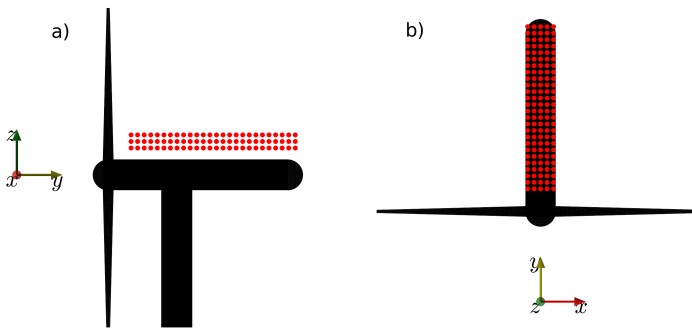

**Figure 2.** Sketch of the setup of the 390 probe positions (in red) from the side view a) and the top view b). The axis origin is located directly in the rotor centre and has been moved in front of the rotor for better visibility.

## 2.4 Free Field Data

Our second approach to verify our hypothesis and find the parameters for our linear model (Eq. (3)) is to examine measurements
from commercial turbines and to carry out experiments in the free field. For this, we used two sets of data. Firstly, measurement data of the BARD 5.0 wind turbine, which is located at the prototype site in north-west Germany on the Rysumer Nacken consisting of two wind turbines of this type. The turbine has a rotor diameter of $122\,\mathrm{m}$, a hub height of $90\,\mathrm{m}$ and a rated power of $5\,\mathrm{MW}$. More details on the BARD 5.0 wind turbine can be found in Teubler (2011). And secondly, data from the eno114 wind turbine from the Kirch Mulsow test field in north Germany. For more information regarding the eno114 and the test field
Kirch Mulsow see (Hulsman et al., 2022).

At both locations, the measurements of the wind vane could be compared with a mast which was set up at a distance of approx. $300\,\mathrm{m}$ in each case. At the eno114, however, we were able to analyse situations within the scope of an investigation into wake deflection, in which the rotor was intentionally misaligned to the wind direction by up to $20°$.

### 2.4.1 Comparison between Wind Vane and Met Mast

In order to identify an error or an overestimation of the wind vane, we compare the measurements of the turbine's wind vane with the wind direction measurements of the meteorological mast (met mast) as a reference.

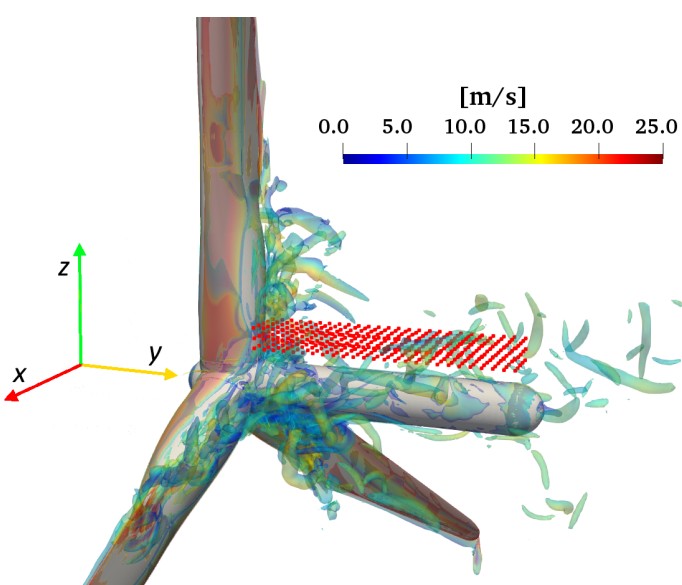

**Figure 3.** Nacelle region of the simulated rotor with the 390 probe locations marked in red. The instantaneous root vortices that influence the probe region are described using an iso-surface of the $\lambda_2$ criterion of Jeong and Hussain (1995) for a value of $\lambda_2 = 100 \frac{1}{\mathrm{s}^2}$ and are coloured by the velocity magnitude.

At the Rysumer Nacken test site there are two wind vanes installed on the BARD5.0 wind turbine at a height of $91\,\mathrm{m}$ approximately $8\,\mathrm{m}$ behind the rotor plane at a lateral distance of $2\,\mathrm{m}$ for the rotor axis and $1.5\,\mathrm{m}$ above the nacelle. One of the type *Wind direction Sensor INDUSTRY* (00.14567.110040) from Lambrecht meteo GmbH and one of the type *Ultrasonic*
*Anemometer 2D compact* (4.387x.xx.xxx) by This Clima. The two system setup serves to increase availability. When both devices are operational, the yaw control uses the mean value of both wind vanes, which is also the measurement we are investigating. The met mast is equipped with a wind vane at a height of $90\,\mathrm{m}$.

At the wind farm in Kirch Mulsow one ultrasonic wind vane of the type *Ultrasonic Anemometer 2D compact* (4.387x.xx.xxx) by Thies Clima is installed on the eno114 wind turbine in $120\,\mathrm{m}$ height approximately $12\,\mathrm{m}$ behind the rotor and $1.5\,\mathrm{m}$ above
the nacelle. The met mast uses a wind vane of the type Wind Direction Transmitter "First Class" (4.3151.00.x1x) from Thies Clima at a height of $112\,\mathrm{m}$.

Analogous to above, we refer to the wind direction deviation measured by the turbine as $\varphi_{\mathrm{wt}}(t) \in [-180°, 180°)$. To compensate for small scale fluctuations we resample the measurements to 60-s averages. We denote the wind direction measured by the met mast with $\omega_{\mathrm{mm}}(t) \in [0°, 360°)$, from this we calculate the wind direction deviation of the wind turbine determined
by the met mast as $\varphi_{\mathrm{mm}}(t) := \omega_{\mathrm{mm}}(t) - \gamma_{\mathrm{wt}}(t)$ and also resample the data to 60-s averages.

In order to identify the influence of the thrust of the rotor on the wind vane on the wind turbine, only situations in which the wind turbine operated at partial load without curtailment were taken into account for the comparison of the measured values.





To quantify the relation between the two different measured values, we use an "Orthogonal Distance Regression" (ODR) (Boggs and Rogers, 1990). This regression method works similarly to an "Ordinary Linear Regression" (OLS). In both methods,

the measured values are transferred as tuples into a coordinate system (scatter plot), in our case $(\varphi_{\mathrm{wt}}(t), \varphi_{\mathrm{mm}}(t))_t$. In an OLS, an affine linear function is determined that minimises the squared distance of the function values to the dependent variable of the data points, which is the second component of the tupel and which is usually plotted on the vertical axis ($y$-axis) of a graph. This is reasonable if the first component of the tupel, which is usually called the predictor variable or independent variable and is usually plotted on the horizontal axis ($x$-axis) of a graph, does not dependent on other factors and has no uncertainty.

One of the most common examples of this is, if the independent variable is the timestamp. However, if the predictor has uncertainties, the slope of the regression will be biased towards zero. This phenomenon is reffered to as "regression dilution" and is extensively discussed in (Frost and Thompson, 2000). Another consequence of this is that an OLS generally cannot be inverted., i.e. if one inverts the tuples (swaps the $x$ and $y$ values) and applies an OLS to the new tuples, the resulting straight line is generally not the inverse function of the original straight line. In contrast, an ODR minimises the squared orthogonal

distances from a regression function to the tuples. The regression line obtained by this method accounts for uncertainties on both the first and the second component of the data points. With an ODR, the variables can be swapped, thereby inverting the regression line. It should be noted, however, that the uncertainties in both measured values are weighted equally. It is possible to obtain a different weighting for the uncertainties of both measured values by stretching or compressing the data on one of the axes of the coordinate system. In our case, however, we assume that the 60-s mean values considered have similar uncertainties.

The gradient of the ODR regression line gives us an estimate for the correction factor $c$. The results of this investigation are presented in section 3.2.1.

### 2.4.2   Yaw Manoeuvre Analysis

In this section, we introduce a method for estimating the correction factor for the wind vane without the need for external measurements, such as a measuring mast. Similar to the step response analysis for time-invariant linear systems, we compare

the wind vane measurements immediately before a yaw manoeuvre with those after.

First, we filter the SCADA data to exclude situations in which the wind turbine is not generating electricity or is operating at reduced output. Then, we identify all yaw manoeuvres in the SCADA data and divide them into clockwise (cw) and counterclockwise (ccw) yaw manoeuvres. On the one hand, the distinction between the two directions of rotation is important for the processing of the data, since the different signs for both directions of rotation must be taken into account in the statistical

evaluations. Another point is that by the distinction a correction factor for both directions of rotation can be determined and so a possible asymmetry can be detected, which can arise for example from an offset error, which was not sufficiently eliminated by the calibration of the wind vane. Regular yaw manoeuvres usually take less than 30 seconds. In some cases, however, a yaw manoeuvre can take longer to perform. This indicates that the wind turbine is realigning itself after a shutdown, that a cable de-twist is taking place or that the wind direction is abruptly changing very strongly. To exclude these cases from our investiga-

tions, we only consider yaw manoeuvres that lasted less than 30 seconds. In the following, we restrict ourselves to describing the methods for the cw yaw manoeuvres, since the methods are used analogously for evaluating the ccw yaw manoeuvres.





The time at which the $i$-th cw yaw manoeuvre ($i \in \mathbb{N}$) starts is denoted as $t_{\mathrm{ys},i} \in \mathbb{R}$ and the time at which it ends $t_{\mathrm{ye},i} \in \mathbb{R}$ (ys and ye denote "yaw start" and "yaw end", respectively). The number of all cw yaw manoeuvres obtained in this way is denoted as $n_{\mathrm{cw}}$. For an empirical analysis of the data, we consider the measurements over a period of the the size $T \in \mathbb{R}$ before to the start of the cw yaw manoeuvre $([t_{\mathrm{ys},i} - T, t_{\mathrm{ys},i}])_i$ and respectively after the end of the yaw manoeuvre $([t_{\mathrm{ye},i}, t_{\mathrm{ye},i} + T])_i$. The length of the time interval $T$ must be selected sufficiently small, depending on the configuration of the yaw controller, so that no further yaw manoeuvres within the time interval interfere with the measurements and the yaw angle is constant before and after the yaw manoeuvre, respectively. $T$ must be selected large enough to suppress turbulence-related measurement noise as far as possible. For our investigations we therefore chose $T = 60\,\mathrm{s}$. The measurements during the yaw manoeuvres $[t_{\mathrm{ys},i}, t_{\mathrm{ye},i}]$ are not considered in the analysis because the duration of the yaw manoeuvres varies and there are additional uncertainties during the rotation of the nacelle. For the selected periods, we now consider the wind direction measurements of the wind turbines. For aggregating the data, we center the measurements around the yaw angle at the end of the respective yaw manoeuvre and thus obtain the following expression:

$$
\overline{\omega}_{\mathrm{wt,cw}}(\tau) = \begin{cases} \frac{1}{n_{\mathrm{cw}}} \sum_{i=1}^{n_{\mathrm{cw}}} \omega_{\mathrm{wt}}(t_{\mathrm{ys},i} + \tau) - \gamma_{\mathrm{wt}}(t_{\mathrm{ye},i}), & \text{for } \tau \in [-T, 0\,\mathrm{s}] \\ \frac{1}{n_{\mathrm{cw}}} \sum_{i=1}^{n_{\mathrm{cw}}} \omega_{\mathrm{wt}}(t_{\mathrm{ye},i} + \tau) - \gamma_{\mathrm{wt}}(t_{\mathrm{ye},i}), & \text{for } \tau \in (0\,\mathrm{s}, T] \end{cases} .
\tag{4}
$$

In a similar way, we also average the yaw angles centered around the yaw angle at the end of the respective yaw manoeuvre:

$$
\overline{\gamma}_{\mathrm{wt,cw}}(\tau) = \begin{cases} \frac{1}{n_{\mathrm{cw}}} \sum_{i=1}^{n_{\mathrm{cw}}} \gamma(t_{\mathrm{ys},i} + \tau) - \gamma(t_{\mathrm{ye},i}) & \text{for } \tau \in [-T, 0\,\mathrm{s}] \\ 0° & \text{for } \tau \in (0\,\mathrm{s}, T] \end{cases} .
\tag{5}
$$

As already mentioned the yaw angle is constant before the yaw manoeuvre $\overline{\gamma}_{\mathrm{wt,cw}}(\tau) = \gamma_{\mathrm{wt,cw}}^{\tau \leq 0}$ for $\tau \leq 0\,\mathrm{s}$, only transitions during the yaw manoeuvre and is constant again after the yaw manoeuvre $\overline{\gamma}_{\mathrm{wt,cw}}(\tau) = \gamma_{\mathrm{wt,cw}}^{\tau > 0} = 0°$ for $\tau > 0\,\mathrm{s}$. The wind direction deviations are aggregated without centering according to:

$$
\overline{\varphi}_{\mathrm{cw}}(\tau) := \begin{cases} \frac{1}{n_{\mathrm{cw}}} \sum_{i=1}^{n_{\mathrm{cw}}} \varphi_{\mathrm{wt}}(t_{\mathrm{ys},i} + \tau), & \text{for } \tau \in [-T, 0\,\mathrm{s}] \\ \frac{1}{n_{\mathrm{cw}}} \sum_{i=1}^{n_{\mathrm{cw}}} \varphi_{\mathrm{wt}}(t_{\mathrm{ye},i} + \tau), & \text{for } \tau \in (0\,\mathrm{s}, T] \end{cases} .
\tag{6}
$$

From this, we can calculate the time-averaged deviation of the mean wind direction $\overline{\varphi}_{\mathrm{cw}}$ before the yaw manoeuvre:

$$
\overline{\overline{\varphi}}_{\mathrm{cw}}^{\tau \leq 0} := \frac{1}{T} \int_{-T}^{0\,\mathrm{s}} \overline{\varphi}_{\mathrm{cw}}(\tau)\mathrm{d}\tau,
\tag{7}
$$

and after the yaw manoeuvre:

$$
\overline{\overline{\varphi}}_{\mathrm{cw}}^{\tau > 0} := \frac{1}{T} \int_{0\,\mathrm{s}}^{T} \overline{\varphi}_{\mathrm{cw}}(\tau)\mathrm{d}\tau.
\tag{8}
$$

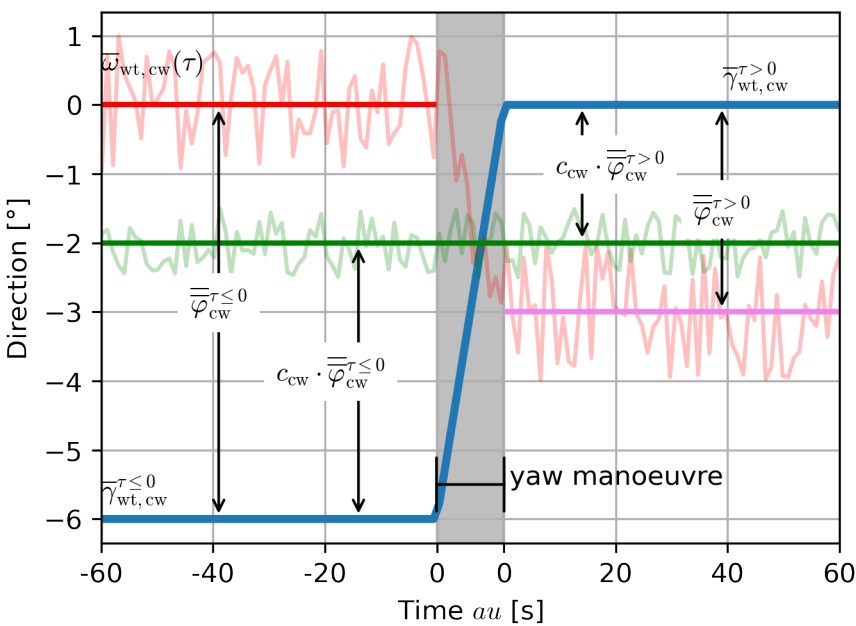

**Figure 4.** Schematic illustration of the average measured wind direction $\overline{\omega}_{\mathrm{wt,cw}}(\tau)$ (in light red) before and after a yaw manoeuvre, the average yaw angle $\overline{\gamma}_{\mathrm{wt,cw}}(\tau)$ (blue) and the average true wind direction $\omega_{\mathrm{ref}}(\tau)$ (green) all centered at the respective yaw angle after the yaw manoeuvre.

Figure 4 gives a schematic illustration of these aggregated measurements. The wind direction $\overline{\omega}_{\mathrm{wt,cw}}(\tau)$ is show in light red together with the average yaw angle $\overline{\gamma}_{\mathrm{wt}}(\tau)$ in blue. The bold red line represents the time-averaged measured wind direction before the yaw manoeuvre, therefore the difference between the bold red line and the yaw angle is $\overline{\overline{\varphi}}_{\mathrm{cw}}^{\tau \leq 0}$ as shown in the figure. Analogously, the bold magenta line represents the time-averaged measured wind direction after the yaw manoeuvre, which means the difference between this value and the yaw angle is $\overline{\overline{\varphi}}_{\mathrm{cw}}^{\tau < 0}$. In green the centered "true" reference wind direction

$\overline{\omega}_{\mathrm{ref}}(\tau)$ is shown. The reference wind direction cannot be measured directly and is there unknown, but we can state the following two assumptions:

First, the measured and the true wind direction deviation can be expressed by the linear function from Eq. (3). The correction $c$ is denoted as $c_{\mathrm{cw}}$, since in this case we only analyse clock-wise yaw manoeuvres. This relation is show in the figure by the difference between the green line and the yaw angle before and the green line and the yaw angle after the yaw manoeuvre,

respectively. Second, the wind direction is a stationary random process for the duration $\tau \in [t_{\mathrm{ys,i}} - T, t_{\mathrm{ye}} + T]$. Now we can postulate the following relationship:

$$\hat{\omega}_{\mathrm{ref,cw}} = \overline{\gamma}_{\mathrm{wt,cw}}^{\tau \leq 0} + c_{\mathrm{cw}} \cdot \overline{\overline{\varphi}}_{\mathrm{cw}}^{\tau \leq 0} = \overline{\gamma}_{\mathrm{wt,cw}}^{\tau > 0} + c_{\mathrm{cw}} \cdot \overline{\overline{\varphi}}_{\mathrm{cw}}^{\tau > 0}. \tag{9}$$





Solving for $c_{\mathrm{cw}}$ gives us:

$$c_{\mathrm{cw}} = \frac{\overline{\gamma}_{\mathrm{wt,cw}}^{\tau>0} - \overline{\gamma}_{\mathrm{wt,cw}}^{\tau\leq0}}{\overline{\overline{\varphi}}_{\mathrm{cw}}^{\tau\leq0} - \overline{\overline{\varphi}}_{\mathrm{cw}}^{\tau>0}}. \tag{10}$$

The yaw control ensures that the yaw angle after the yaw manoeuvre $\gamma_{\mathrm{wt,cw}}^{\tau>0}$ corresponds to the measured wind direction before the yaw $\omega_{\mathrm{wt,cw}}^{\tau\leq0}$ (see Eq. (1)). This holds true especially for the aggregated averages considered here, therefore: $\overline{\gamma}_{\mathrm{wt,cw}}^{\tau>0} = \overline{\omega}_{\mathrm{wt,cw}}^{\tau\leq0} = \overline{\gamma}_{\mathrm{wt,cw}}^{\tau\leq0} + \overline{\overline{\varphi}}_{\mathrm{cw}}^{\tau\leq0}$. The correction factor can thus be estimated only by the wind vane measurements:

$$c_{\mathrm{cw}} := \frac{\overline{\overline{\varphi}}_{\mathrm{cw}}^{\tau\leq0}}{\overline{\overline{\varphi}}_{\mathrm{cw}}^{\tau\leq0} - \overline{\overline{\varphi}}_{\mathrm{cw}}^{\tau>0}}. \tag{11}$$

We proceed analogously for ccw yaw manoeuvres.

## 3 Results

In this section the results of the methods described in the previous section are presented. Since multiple different datasets were analyzed for these studies there is a short overview here:

- In Section 3.1, the data from the CFD simulation is analyzed as described in Section 2.3.

- In the Section 3.2, the methods described in Section 2.4 are applied to measured data from the free field.

    - In Subsection 3.2.1 the wind wane measurements from the BARD5.0 at the Rysumer Nacken and the eno114 at Kirch Mulsow are compared to wind direction measurements form the met mast at their respective locations as described in Section 2.4.1 .

    - In Subsection 3.2.2 the yaw step analysis described in Section 2.4.2 is applied to the BARD5.0 wind turbine at Rysumer Nacken.

    - And finally 3.3.1 shows results from a free field experiment where the correction factor was applied to the yaw controller of the BARD5.0 wind turbine at Rysumer Nacken.

### 3.1 CFD Simulation

From the data generated by the CFD simulation we omitted the first 10 seconds of the simulation in our evaluation, in which the wake develops directly behind the rotor. In addition, we only consider measuring points at a minimum distance of $5\,\mathrm{m}$ behind the rotor plane, which means that 300 of the 390 probes were used for the evaluation. The measured values were averaged over the three different measurement heights since the observation of the individual measurement heights did not reveal any special features. Figures 5 a) to e) show the time-averaged wind directions for the different measurement points behind the rotor in colour.



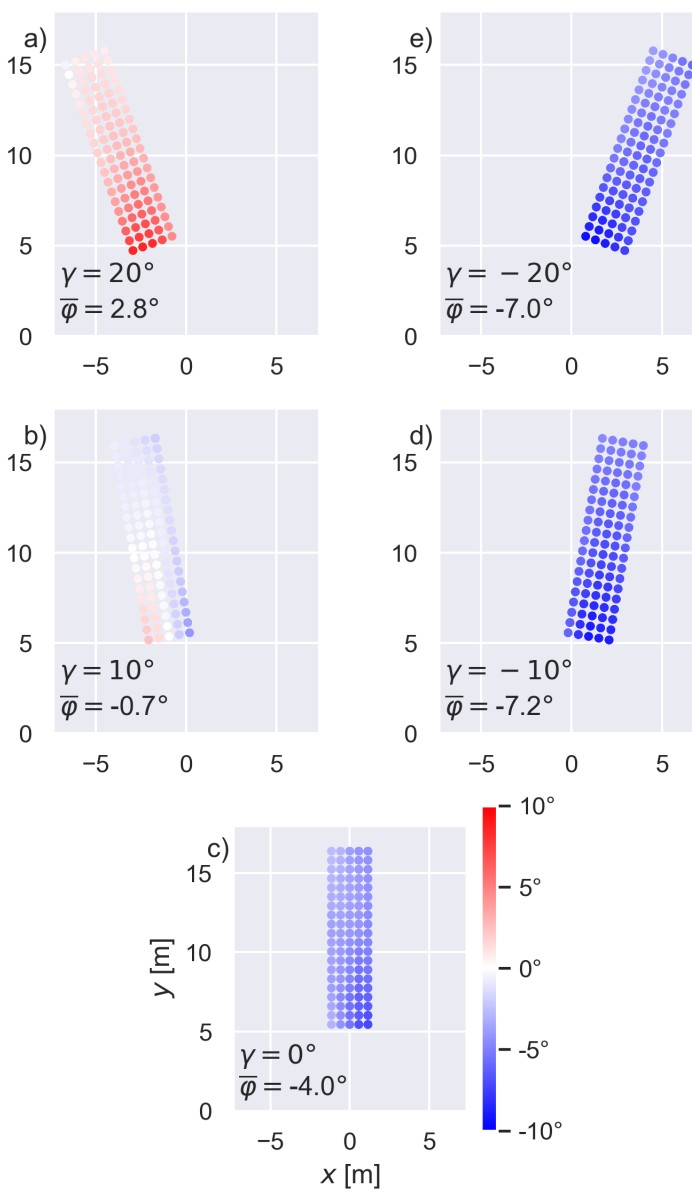

**Figure 5.** Coloured representation of the averaged wind direction above the nacelle for five different misalignments $\gamma \in \{-20°, -10°, 0°, 10°, 20°\}$. A red colouring represents a deflection to the right (positive) and a blue colouring a deflection to the left (negative). The mean wind direction of all measurement points over the whole simulation time is given as $\overline{\varphi}$.

The plots display the deviations of the wind directions from the incoming wind direction coming from the south. A red
colour indicates a deflection to the right (positive direction) and a blue colour to the left (negativ direction). In Figure 5 a)



the wind turbine is yawed by $20°$ (counterclockwise) relative to the inflow. The flow at the measurement locations is relative homogeneous, with a stronger deviation closer to the rotorplane. The average flow at the measurement locations has a direction of $\overline{\varphi} = 2.8°$. The total average misalignment a virtual wind vane would experience is therefore $22.8°$. Figure 5 b) depicts the situation where the wind turbine is yawed by $10°$ (counterclockwise) relative to the inflow. The flow at the measurement

locations shows very little deviation from the inflow. A small positive trend on the right hand side of the figure close to the rotor plane and a negative trend on the left hand side of the figure can be observed. The overall average of the flow at the measurement locations is $\overline{\varphi} = -0.7°$, so the total misalignment the virtual wind vane would measure is $9.3°$. In Figure 5 c) the wind turbine is aligned with the inflow. At the measurement locations a deviation to the left with an average of $\overline{\varphi} = -4.0°$ can be seen. This shows the aforementioned deviation due to the rotation of the blades, which cause a counter-rotation of the

flow behind the rotor plane. In Figures 5 d) where the wind turbine is misaligned by $-10°$ (i.e. $10°$ in clockwise direction), this deflection gets even stronger with a average flow direction of $\overline{\varphi} = -7.2°$. A virtual wind vane would experience on average a misalignment of $-17.2°$. Similarly in the last Figure 5 e) the misalignment is comparable to the case before, with an average misalignment of $\overline{\varphi} = -7.0°$. A virtual wind vane would experience a misalignment of $-27.0°$.

Figure 6 summarizes the average virtual measured misalignments (blue dots), additionally a ordinary least square (OLS)

regression was fitted to the measurements (red line), that shows a very strong linear trend with a correlation coefficient of $R = 0.998$. Here it can be observed that at the measuring points an amplification of the actual misalignment is measured in the magnitude of $26\%$ (the slope is $m = 1.26$). To compensate this amplification factor by a correction factor this factor must be $c = \frac{1}{m} = 0.79$.

The offset of the regression is $b = 3.20°$. Please note, we mentioned in Section 2.2 that the offset for the correction is

assumed to be zero when using real measured data, since such an offset should be accounted for by the calibration of the wind vane. For the data from the simulation, there is neither a real wind vane nor a calibration and therefore we also get the offset in the numerical simulation results.

## 3.2 Free Field Measurements

### 3.2.1 Comparison Wind Vane versus Met Mast

For the evaluation of the measurements in the free field, we compare the wind direction measurements on the wind turbine nacelle with the measurements at the met mast, as described in Section 2.4.1. First we present the results from the test field Rysumer Nacken introduced in Section 2.4.1. Here, we only considered wind directions between $180°$ and $360°$, as the met mast is in free flow in this wind direction range and is not disturbed by wakes. The measurements were recorded in the period from 12.09.2020 to 08.10.2020. After filtering, the amount of viable 60-s measurements is $n = 8013$. In Figure 7, a 2D histogram of

yaw offset measured by the wind vane at the BARD5.0 wind turbine compared to the yaw offset measured met mast is shown.

The black contour lines show a kernel density estimate and the red line is the result of the ODR (see Section 2.4.1). The slope of the regression is, which also provides the correction factor is $c := m = 0.83$. The offset is $b = -1.91°$, this offset can be a result of slightly different northings between the wind turbine and the met mast.

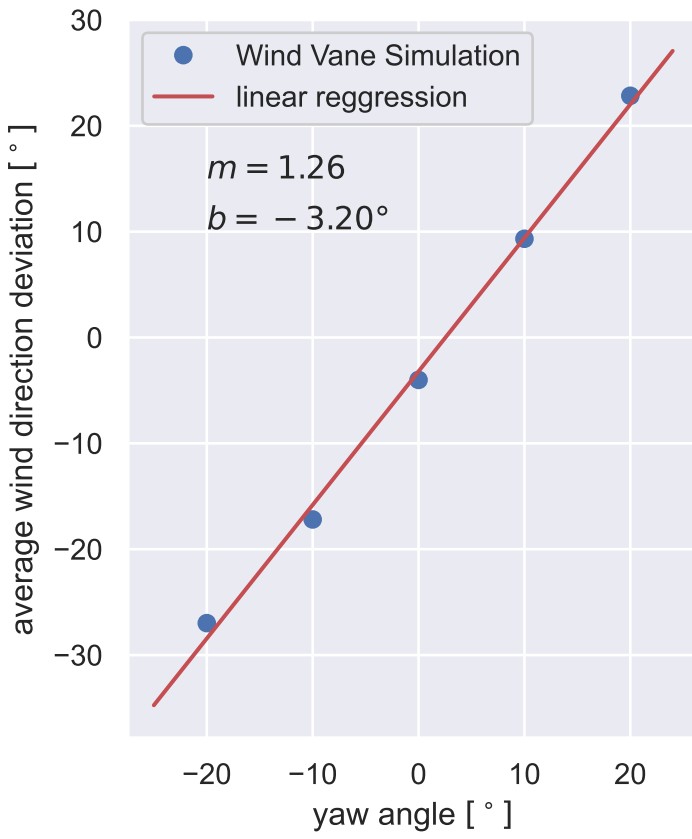

**Figure 6.** The blue dots represent the average deflection above the nacelle for the five different misalignments from the perspective of the nacelle longitudinal axis, that were simulated in the CFD. In red, a linear regression line was fitted through the points. The slope $m$ and the axis intercept point $b$ are stated.

Next, we show the results from the test field Kirch Mulsow. In this small wind farm, we only considered wind directions from $215°$ to $300°$, which is the sector with free inflow for the met mast, for this evaluation. The measurements were recorded in the perion from 19.01.2021 to 03.07.2021. The amount of viable measurements after filtering is $n = 45042$. Figure 8 shows the comparison of the measured values of the met mast at the Kirch Mulsow site with those of the wind vane at the eno114 wind turbine in the same way as before. The ODR in this case results in a correction factor of $c := m = 0.80$ and the offset is $b = 0.49$.

Since intentional misalignments of the rotor of up to $20°$ in both directions were also tested at this test site, our range of values here extends much further than in the first case.

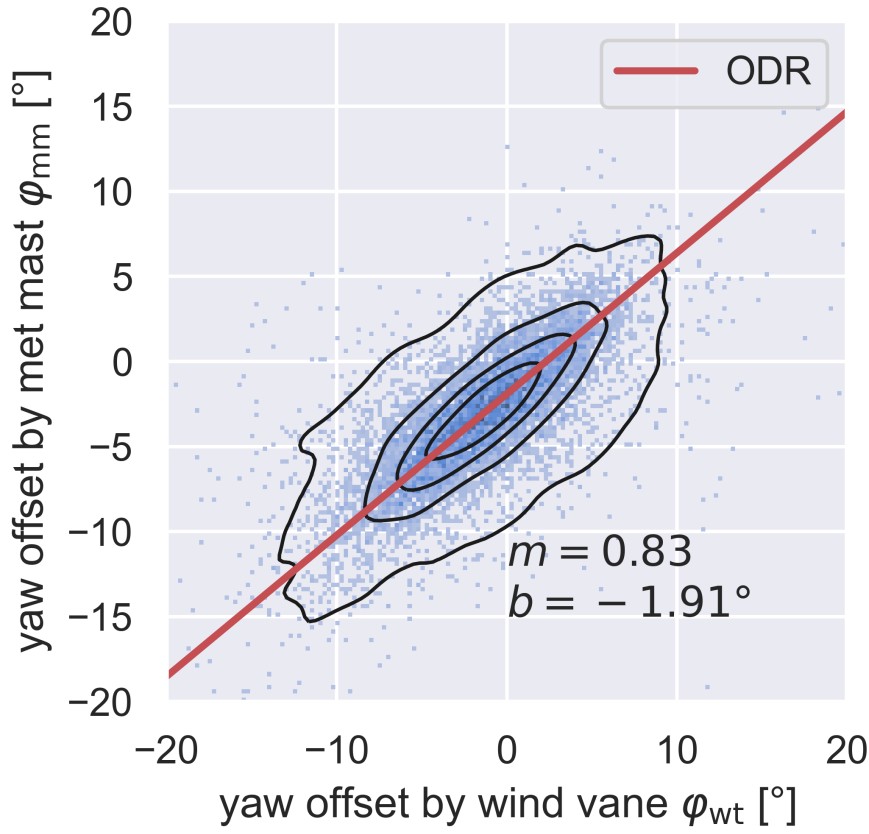

**Figure 7.** 2D histogram of the yaw misalignment determined by the wind vane at the BARD5.0 $\varphi_{\mathrm{wt}}$ ($x$-axis) and the met mast $\varphi_{\mathrm{mm}}$ ($y$-axis) at the Rysumer Nacken. Kernel density estimation is shown in black contour lines and the ODR is displayed in red.

### 3.2.2 Wind Vane Measurements before and after Yaw Actuation

In this section, an analysis of wind direction measurements by the wind vane at the BARD5.0 wind turbine in the Rysumer Nacken test area during the period from 01.08.2020 to 08.10.2020 is performed using the methods presented in section 2.4.2.

g averages. Since the yaw manoeuvre can have an influence on the measured values due to the time averaging, we excluded the measurements 10 seconds before and after the yaw manoeuvre from the analysis. Figure 9 shows the 2D histograms of the wind vane measurements before and after the yaw manoeuvre in cw direction, and Figure 10 the same for yaw manoeuvres in ccw direction.

In this time period, we recorded 4234 yaw manoeuvres in cw direction and 4082 in ccw direction.

The time series of each yaw manoeuvres were averaged as described in Eq. 4 and plotted as a black graph, with the 99 % confidence interval. The individual measurements of the wind direction are displayed in a 2D histogram, which reflects the frequency by a blue coloration. The yaw angle was averaged according to Eq. 5 of the wind turbine is displayed in white.

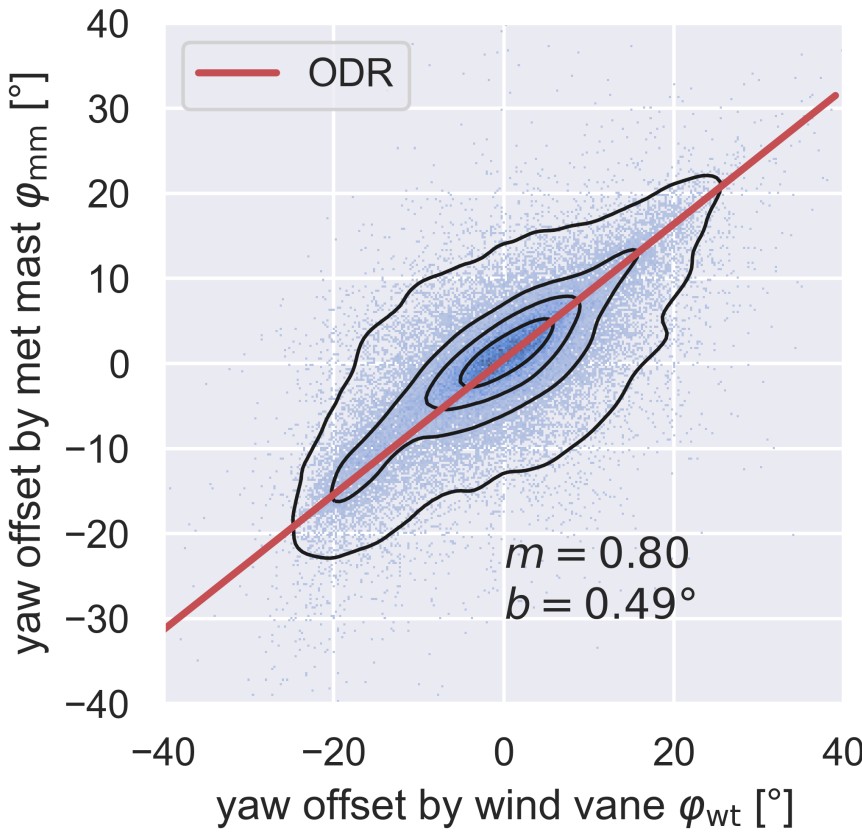

**Figure 8.** 2D histogram of the yaw misalignment determined by the wind vane at the eno114 $\varphi_{\text{wt}}$ (x-axis) and the met mast $\varphi_{\text{mm}}$ (y-axis) at the Kirch Mulsow site. Kernel density estimation is shown in black contour lines and the ODR is displayed in red.

Since the yaw manoeuvres can be of different magnitudes, we have also given a confidence range for the yaw angle before the manoeuvre, although the confidence range is very small, so that it is hardly visible.

In both figures, an increase in the wind direction deviation can be seen before the yaw manoeuvre. This increase of the deviation can be explained by the fact that a yaw manoeuvre is triggered by the yaw controller when a moving average value of the wind direction deviation exceeds a certain threshold value. Since we filtered for exactly these situations, we can see an increase of the moving average up to the point where a yaw manoeuvre is triggered.

    Both figures further reveal that after a yaw manoeuvre, on average, the measured wind direction does not match the orienta-

tion of the wind turbine, but that the wind turbine has overshot the target by 2° to 3° for both cw and ccw yaw directions.

    For the calculation of the correction factors we are using Eq. 11. From the measurements of the cw-yaw manoeuvres we retrieve an average wind direction deviation before the yaw manoeuvre of $\overline{\overline{\varphi}}_{\text{cw}}^{\tau \leq 0} \approx 9.23°$ and after the yaw manoeuvre of $\overline{\overline{\varphi}}_{\text{cw}}^{\tau > 0} \approx -2.11°$, which results in a correction factor of $c_{\text{cw}} = \frac{\overline{\overline{\varphi}}_{\text{cw}}^{\tau \leq 0}}{\overline{\overline{\varphi}}_{\text{cw}}^{\tau \leq 0} - \overline{\overline{\varphi}}_{\text{cw}}^{\tau > 0}} \approx \frac{9.23°}{11.34°} \approx 0.81$. Analoguously, the wind direction





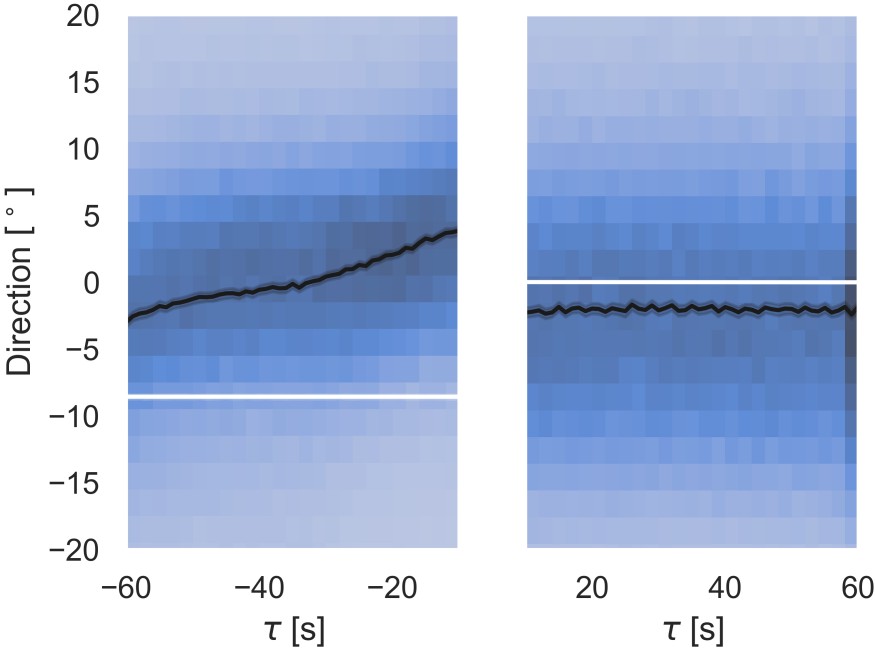

**Figure 9.** 2D histogram of wind direction measured by the wind vane of the BARD5.0 wind turbine before (left) and after (right) a cw-yaw manoeuvre centered around the yaw angle at the end of the yaw manoeuvre. The blue colouring indicates the number of occurrences, with a darker blue indicating a higher count. The average wind direction $\overline{\omega}$ (see Eq. 4) is shown in black with a 99 % confidence interval displayed by the thin grey band around the black line. The yaw angle $\overline{\gamma}$ (see Eq. 5), also with it's 99 % confidence interval, is displayed in white.

deviation before the yaw manoeuvre for the ccw-cases is $\overline{\overline{\varphi}}_{\mathrm{ccw}}^{\tau \leq 0} \approx -8.28°$ and after the yaw manoeuvre $\overline{\overline{\varphi}}_{\mathrm{ccw}}^{\tau > 0} \approx 2.11$. The

correction factor for the ccw-yaw manoeuvres therefor is $c_{\mathrm{ccw}} = \frac{\overline{\overline{\varphi}}_{\mathrm{ccw}}^{\tau \leq 0}}{\overline{\overline{\varphi}}_{\mathrm{ccw}}^{\tau \leq 0} - \overline{\overline{\varphi}}_{\mathrm{ccw}}^{\tau > 0}} \approx \frac{-8.28°}{-10.39} \approx 0.80$.

### 3.3 Free Field Experiment of Wind Vane Correction

We conducted experiments in the test field Rysumer Nacken on the BARD5.0 wind turbine to investigate the effects of wind vane correction on the operation of a commercial wind turbine. Since the operation of a wind turbine is dependent on strong uncontrollable and random conditions, we wanted to build a database that would give us a good comparison between normal

operation and operation with wind vane correction. For this reason, we ran the wind turbine alternately for one hour in normal operation and one hour with wind vane correction enabled, and repeated this procedure until we collected a sufficient amount of data. We refer to this procedure as a "toggle test".

We performed the first toggle test in the period from 06.07.2021 to 26.08.2021. In this experiment, we use the correction model described above (see Eq. (11)) with a correction factor of $c = 0.8$ for the wind vane, since our investigations at the time

of the experiments resulted in this correction factor. This means that the wind vane signal $\varphi_{\mathrm{wt}}(t)$, which is used in the yaw



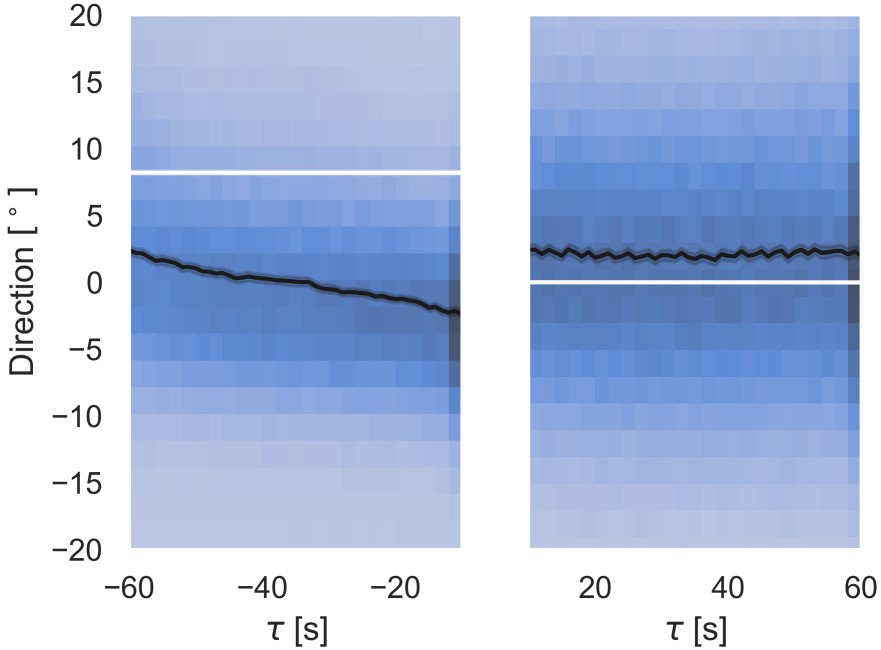

**Figure 10.** 2D histogram of wind direction measured by the wind vane of the BARD5.0 wind turbine before (left) and after (right) a ccw-yaw manoeuvre centered around the yaw angle at the end of the yaw manoeuvre. The blue colouring indicates the number of occurrences, with a darker blue indicating a higher count. The average wind direction $\overline{\omega}$ (see Eq. 4) is shown in black with a 99 % confidence interval displayed by the thin grey band around the black line. The yaw angle $\overline{\gamma}$ (see 5), also with it's 99 % confidence interval, is displayed in white.

controller to steer the wind turbine orientation, was multiplied by this factor directly at the input i.e. for the yaw trigger as well as the yaw target. We denote the corrected wind vane measurement as $\varphi_{\mathrm{corr}}(t) = c \cdot \varphi_{\mathrm{wt}}(t)$.

In the second toggle test, which was performed in the period from 01.09.2021 to 19.11.2021, we also use a correction factor of $c = 0.8$. In this test, however, the unmodified wind vane measurement $\varphi_{\mathrm{wt}}(t)$ is used to for the yaw trigger. Only for the
yaw target the corrected value $\varphi_{\mathrm{corr}}(t)$ is applied.

For the evaluations of the tests, we consider the number of yaw manoeuvres in the respective test period, we analyze the "step response" of the measured wind direction deviation during a yaw manoeuvre according to section 2.4.2, and we investigate the influence on the power output by calculating the changes in the power curve.

### 3.3.1  Evaluation of the Toggle Tests

This section summarizes the results of the two toggle tests that were conducted in the Rysumer Nacken test site at the BARD5.0 wind turbine. Table 1 lists the most important statistics of the toggle tests, which are referred to in the following when describing the individual test results.



**Table 1.** Results of the toggle tests at Rysumer Nacken.

| statistic | Toggle Test 1 | | Toggle Test 2 | |
|---|---|---|---|---|
| | Normal operation | WVane Correction | Normal operation | WVane Correction |
| total duration | 21.07 days | 20.95 days | 31.93 days | 32.52 days |
| number of yaw actuations | 4960 | 2869 | 5373 | 5047 |
| yaw actuations per 10 min | 1.64 | 0.95 | 1.17 | 1.08 |
| total yaw distance | 41450° | 21954° | 42625° | 31522° |
| yaw distance per 10 min | 13.66° | 7.28° | 9.27° | 6.73° |

Even though the period of normal operation was the same as the period with activated wind vane correction when performing a toggle test, the "total durations" differ from each other here. This can be explained by the data filtering, since only periods were considered in which the wind turbine was active, operated in the partial load range, and no active power curtailment was applied.

In the first toggle test, the turbine performed on average $0.95$ yaw actuations in 10 minutes with the activated wind vane correction, turning $7.28°$ per 10 min on average. Compared to $1.64$ yaw manoeuvres and a yaw distance of $13.66°$ per 10 min on average in normal operation this is a reduction of $41.8\,\%$ in actuations and $46.7\,\%$ in total yaw distance.

In the second toggle test the yawing activity was reduced from $1.169$ yaw actuations per 10 minutes in regular operation to $1.078$ per 10 minutes with wind vane correction activated, i.e. by $7.8\,\%$. The yaw distance experienced a reduction of $27.4\,\%$ from $9.27°$ to $6.73°$ per 10 minutes due to the wind vane correction.

For both test cases, we aggregated the wind direction deviation before and after the yaw processes as described in Section 2.4.2 and present them in Figures 11 and 12 for the first test case analog to Figure 9 and Figure 10 . The measurements shown are the 10-s moving average wind direction measurements. It can be seen that the alignment of the wind turbine and the measured wind direction at the wind vane match better after the yaw process compared to Figure 9 and Figure 10. The result for the second test case looks very similar to this figure, so we do not show it here.

To identify the influence of the wind vane correction on the performance, we determined the power curves binned in 1 m/s steps from the 10-min data for both conditions in both test cases and calculated the absolute power difference for each bin $P_\text{diff} = P_\text{corr} - P_\text{standard}$, where $P_\text{corr}$ is the power output with the wind vane correction active and $P_\text{standard}$ is the power output during standard yaw control, with no wind vane correction.

Figure 13 shows the difference of these power curves for the first toggle test and Figure 14 for the second toggle test. In addition to the difference, we have added error bars. These were calculated from the square root of the sum of the squared standard errors of the mean of both power curves multiplied by 2.576 to give an estimate of the $99\,\%$ confidence interval.

Figure 13 shows that in the first toggle test on average less power is produced by the wind vane correction for most wind speeds, but the fluctuations in the measurements are so large that this is not statistically significant. In Figure 14, more power is

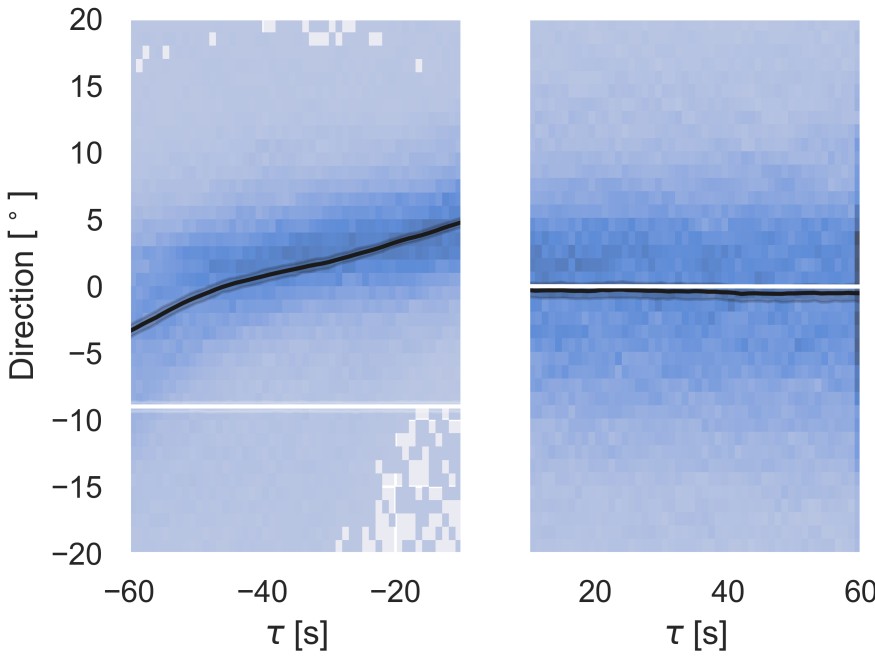

**Figure 11.** 2D Histogram of wind direction measured by the wind vane of the BARD5.0 before (left) and after (right) a cw-yaw manoeuvre centered around the yaw angle at the end of the yaw manoeuvre for activated wind vane correction in the first toggle test. The average wind direction and its 99 % confidence interval is shown in black. The yaw angle is displayed in white.

produced during the second toggle test for most wind speeds by the wind vane correction, but the difference is not statistically significant here either.

For both test cases we calculated the influence on the Annual Energy Production (AEP) using the average values and an assumed Weibull wind distribution (Weibull scale parameter $A = 11.33\,\mathrm{m/s}$ and Weibull shape parameter $k = 2.29$). For toggle test 1, this results in a loss of power generation of $-0.43\,\%$ and for toggle test 2 an increase in power of $0.06\,\%$.

## 4 Discussion

The CFD simulations we performed (see Section 3.1) to better understand the mean wind direction immediately behind the rotor plane, support our hypothesis that the rotor's thrust deflects the flow at the wind vane location during yaw misalignment. This effect affects the far wake, as previously shown in other studies (Jiménez et al., 2010; Bastankhah and Porté-Agel, 2016), and has an impact on the flow directly above the nacelle, and thus, on conventional wind vanes of a wind turbine. However, our simulations only serve as a proof-of-concept, as we used ideal conditions (uniform flow) for the calculations in order to be able to represent the effect of the rotor on the flow in isolation. We focused our investigation on an average error in the wind vane signal. For modelling the wind vane error, we have assumed a simple affine linear function, which seems to be confirmed in

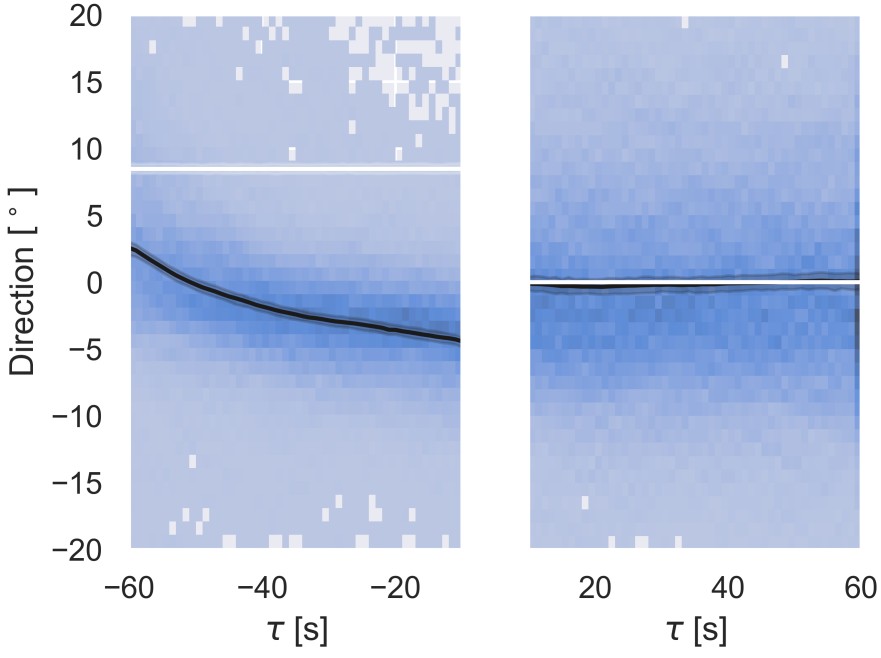

**Figure 12.** 2D Histogram of wind direction measured by the wind vane of the BARD5.0 before (left) and after (right) a ccw-yaw manoeuvre centered around the yaw angle at the end of the yaw manoeuvre for activated wind vane correction in the first toggle test. The average yaw misalignment and the 99 % confidence interval is shown in black. The yaw angle is displayed in white.

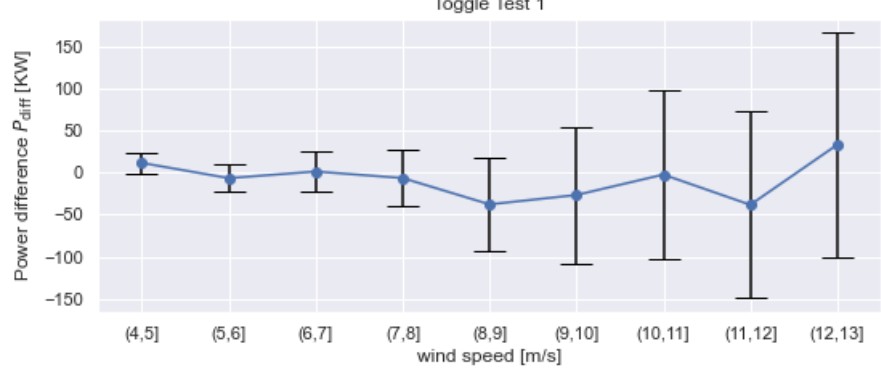

**Figure 13.** Power difference between activated wind vane correction and regular operation over wind speed binnend in $1\,\mathrm{m/s}$, with error bar, which represent the 99 % confidence interval for Toggle Test 1.

the CFD simulations in the range of $-20°$ to $20°$. In general, we believe that this model is appropriate for small misalignments but no longer applies for larger misalignments, as the turbine's thrust decreases in these situations and the overestimation of





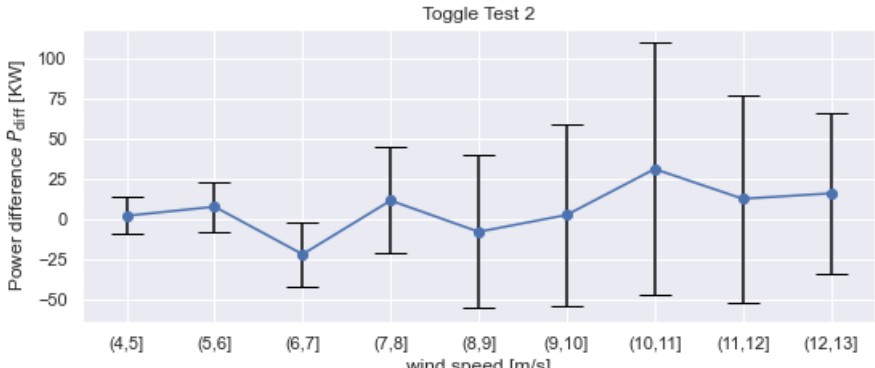

**Figure 14.** Power difference between activated wind vane correction and regular operation over wind speed binnend in $1\,\mathrm{m/s}$, with error bar, which represent the $99\,\%$ confidence interval for Toggle Test 2.

the wind vane should not increase linearly anymore. The influence of more complex conditions, such as increased turbulence intensity as in an unstable atmospheric condition or stronger veer and shear in the flow as in a stable atmospheric condition, was not investigated in the simulation here. For further future investigations, spatially high-resolution LES simulations could
be used to study complex conditions and dynamic inflows in combination with different yaw controllers.

The comparisons between met mast measurements and wind turbine measurements (see Section 3.2.1) also show that the wind vane tends to overestimate the deviation of the wind direction, both at the BARD5.0 wind turbine at the Rysumer Nacken test site and at the eno114 wind turbine in Kirch Mulsow. A direct comparison is, of course, complex, as the met mast is located at a distance from the wind turbine and therefore experiences a slightly different flow. In addition, we have filtered out wind
direction sectors for each site, as in these, the met mast would be in the wake of the surrounding wind turbines, and the wind direction measurement would thus be affected. Nevertheless, our hypothesis is confirmed on average by the measurements on both test sites. In the case of the Rysumer Nacken test field, it can be observed that the measurement data is not centred around the $0°$. This is due to the fact that the measurement values of the wind vane are the raw values without the offset correction and that the measurement mast has a slightly different northing to the orientation of the wind turbine. At the Kirch
Mulsow test site, we received the already offset-corrected measured values. However, in the case of Kirch Mulsow it is also interesting that we have considered larger misalignments here and it shows in Figure 8 that the values in the centre apparently receive a proportionally larger deflection than the outer values, which speaks against a linear model for the overestimation. One can assume that the thrust decreases with greater misalignment, and therefore the deflection also becomes weaker and no longer increases linearly. For the misalignments analysed here, however, the linear model seems to be sufficiently accurate.
In a similar study, Simley et al. (2021) compared wind vane measurements with those obtained from a nacelle-based lidar, in contrast to our study which utilized measurements from the met mast. Although Simley conducted his study on a different turbine and employed different reference measurements, his findings are consistent with our own, demonstrating that the wind vane overestimates wind direction deviation.





Simley also encountered a problem with regression dilution when performing linear regression. To mitigate this issue, he
attempted to reduce the uncertainties in the prediction variable through binning. In contrast, we opted to use orthogonal distance
regression (ODR), which accounts for the uncertainties in both variables, to circumvent the problem with linear regression.

The comparison of the wind vane measurements before and after the yaw manoeuvre (Section 3.2.2) has the advantage that
it can be performed without external measurements, and thus a correction factor for the wind vane can be determined for a
wind turbine that has no reference measurement, such as a measuring mast or a lidar. Similar to the analysis of a step response
for linear time-invariant control systems, this method opens up the possibility of analysing the effect of misalignment on a
variety of variables, such as power, wind speed measurements or load measurements, if available, albeit in an empirical rather
than deterministic manner.

The evaluations of both toggle tests (Section 3.3.1) have shown that the correction factor improved the wind turbine's
alignment in the wind direction on average after the yawing process. In the first toggle test, the correction factor was applied
to the wind vane measurement for the yaw trigger as well as the yaw target. The results show, that the yawing activity was
significantly reduced (by 41.82 %), but this was at the expense of performance, as the corrected wind vane resulted in larger
misalignments than in normal operation. However, the reduction of yawing activity is not only related to allowing larger
misalignments, as shown by the evaluation of the second toggle test, where the yaw trigger is the same as in normal operation,
only the yaw target is affected by the corrected wind vane signal. Nevertheless, the yawing activity was still reduced by 7.78 %
compared to normal operation due to a better alignment of the wind turbine after the yaw manoeuvre. Especially the number
of alternating yaw manoeuvres could be significantly reduced. And since the yaw distance of each yaw manoeuvre was by
definition 20 % shorter in the second toggle test, the total yaw distance was reduced by more than 27 %. The performance
could even slightly increase in the second toggle test due to the better alignment. Although both the decrease in power on
the first toggle test and the increase in power on the second toggle test are not statistically significant. Longer test periods are
needed to determine this. Especially because the measurements are not independent samples but correlated time series. The
effective sample size is therefore even smaller.

Table 1 reveals that the time period of the tests has a large influence on the number of yaw manoeuvres. We suspect that
during the second toggle test, which took place in autumn, stable atmospheric stratification was more frequent compared to
the first toggle test in summer, and that this had an influence on the number of yaw manoeuvres in general. Therefore, it made
sense to conduct toggle tests, as the conditions for both test states were largely identical. Nevertheless, a longer test period
and a differentiation of the effects of the wind vane correction for different atmospheric stabilities could provide further useful
insights.

In our study, each of our investigations (CFD simulation in Section 3.1, wind vane to met mast comparison in Section
3.2.1, yaw manoeuvre analysis (Section 3.2.2)) resulted in a correction factor of approximately 0.8 and the toggle tests in
Section 3.3.1 shows great improvement for applying a wind vane correction with this factor. This indicates the general order
of magnitude for the correction factor, but this value will depend on the shape of the turbine nacelle and the placement of the
wind vane on the nacelle. Therefore, the authors advise that the methods presented here be carefully repeated using data from
a wind turbine before applying a correction factor to other wind turbine types.

Overall, our results are consistent with those of (Simley et al., 2021) and (Kragh and Fleming, 2012), indicating that for yaw
control based on wind vanes, transfer functions in addition to offset calibration are required to correct wind vane overshoot.
In our investigation, we presented methods for parameterizing the transfer function for the specific wind turbine. Atmospheric
stability can have a significant impact on these parameters. Therefore, in future studies, we will conduct analyses to determine
atmospheric stability and filter data accordingly.

## 5   Conclusions

Our study on wind vane measurements during yaw misalignment on two commercial wind turbines and a CFD simulation
revealed that wind direction deviation was overestimated by about $20\,\%$ to $30\,\%$, with the CFD simulation supporting the
hypothesis that this is an inherent characteristic due to the rotor's thrust. To mitigate this problem, we developed a linear
correction function and two data-driven methods to parameterize it. These methods involved using measurements from a
meteorological mast or analyzing wind direction measurements before and after yaw manoeuvres. We tested the correction
function on one wind turbine in two scenarios. In the first scenario, the correction was applied to both the yaw trigger, resulting
in a reduction of yaw activity by more than 40 %. In the second scenario, the correction was only applied to the yaw target,
resulting in a reduction of yaw activity by approximately 8 %. Results also indicated an improvement in alignment with the
flow in both scenarios, while power production of the wind turbine was not significantly influenced. Our finding suggest that a
corrected wind vane signal is crucial for improved wind turbine control strategies, particularly for wake deflection strategies.
Future studies could explore the use of our correction function and data-driven methods on other wind turbine types and in
different environmental conditions.

*Author contributions.*   AR coordinated the research, developed the methods, and performed the analysis of the measured data. LH performed
the CFD simulation, provided the simulation data, and assisted with valuable discussions. PH aided with helpful discussions and supported
in gaining access to and an overview of the data from the Kirch Mulsow site. LJL provided important advice regarding the simulations
and provided thorough internal review and valuable feedback. CM was responsible for providing data from the Rysumer Nacken test site,
integrating the correction factor into the BARD5.0 turbine controller, performing the toggle tests, and providing important advice on methods
and data analysis. MK provided thorough internal review and valuable feedback and had a supervisory function.

*Competing interests.*   The authors declare that they have no conflict of interest.

*Acknowledgements.*   This work was partially funded by the German Federal Ministry for Economic Affairs and Climate Action (BMWK) in
the scope of the project YawDyn (FKZ 03EE3019) in addition, data from the project CompactWind II (FKZ 0325492) were used. We thank





Ocean Breeze Energy GmbH & Co. KG and eno energy systems GmbH for the access to wind turbines and measurement data and fruitful discussions.

## Nomenclature

$\gamma_{\mathrm{wt,cw}}^{\tau>0}$    Yaw angle after the yaw manoeuvre centered around the yaw angle at the end of the yaw manoeuvre averaged over all considered yaw manoeuvres, i.e. $\gamma_{\mathrm{wt,cw}}^{\tau>0} = 0$

$\gamma_{\mathrm{wt,cw}}^{\tau\leq0}$    Yaw angle before the yaw manoeuvre centered around the yaw angle after the yaw manoeuvre averaged over all considered cw yaw manoeuvres

$\gamma_{\mathrm{wt}}$    yaw angle of the wind turbine

$\overline{\gamma}_{\mathrm{wt,cw}}$    Yaw angle centered around the yaw angle at the end of the yaw manoeuvre averaged over all considered cw yaw manoeuvres

$\hat{\varphi}_{\mathrm{ref}}$    Estimate for the reference wind direction

$\overline{\varphi}_{\mathrm{cw}}$    wind direction deviation measured by the wind turbine averaged over all considered cw yaw manoeuvres

$\overline{\overline{\varphi}}_{\mathrm{cw}}^{\tau>0}$    Time average of $\overline{\varphi}_{\mathrm{cw}}$ over the time $T$ after the yaw manoeuvre

$\overline{\overline{\varphi}}_{\mathrm{cw}}^{\tau\leq0}$    Time average of $\overline{\varphi}_{\mathrm{cw}}$ over the time $T$ before the yaw manoeuvre

$\varphi_{\mathrm{corr}}$    Corrected wind vane measurement

$\varphi_{\mathrm{mm}}$    wind direction deviation of the wind turbine to the inflow measured by the met mast

$\varphi_{\mathrm{ref}}$    True or reference wind direction deviation

$\varphi_{\mathrm{wt}}$    wind direction measurement by the wind vane of the wind turbine

$\hat{\omega}_{\mathrm{ref,cw}}$    Estimate of the time averaged wind direction before and after the cw yaw manoeuvre centered around the yaw angle at the end of the cw yaw manoeuvre

$\omega_{\mathrm{mm}}$    global wind direction measured by the met mast

$\omega_{\mathrm{wt}}$    wind direction in the global frame of reference measured by the wind turbine

$\overline{\omega}_{\mathrm{wt,cw}}$    wind direction centered around the yaw angle at the end of the respective yaw manoeuvre, averaged over all considered cw yaw manoeuvres

$\tilde{\omega}_{\mathrm{wt}}$    Moving time averaged global wind direction measured by the wind turbine

$A$    Weibull scale parameter





| | |
|---|---|
| $b$ | Offset of the transfer function |
| $c$ | wind vane correction factor and slope of the transfer function |
| $c_{\mathrm{cw}}$ | Correction factor estimated by the yaw manoeuvre analysis for cw yaw manoeuvres |
| $D$ | Diameter of the wind turbine |
| $i$ | index variable for the number of cw/ccw yaw manoeuvres |
| $k$ | Weibull shape parameter |
| $n_{\mathrm{ccw}}$ | Number of ccw yaw manoeuvres |
| $n_{\mathrm{cw}}$ | Number of cw yaw manoeuvres |
| $P_{\mathrm{corr}}$ | Average power output of the wind turbine with the corrected wind vane signal over the wind speed |
| $P_{\mathrm{diff}}$ | Absolute power difference between $P\mathrm{corr}$ and $P_{\mathrm{standard}}$ |
| $P_{\mathrm{standard}}$ | Average power output of the wind turbine over the wind speed with standard yaw control |
| $T$ | Size of time interval before and after yaw manoeuvre considered. |
| $t$ | Time |
| $t_{\mathrm{ye},i}$ | Time at which the $i$-th cw/ccw yaw manoeuvre ends |
| $t_{\mathrm{ys},i}$ | Time at which the $i$-th cw/ccw yaw manoeuvre starts |



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
