# Peer review of "Wind Vane Correction during Yaw Misalignment for Horizontal Axis Wind Turbines"

_Wind Energy Science, 2023_

## Author Comment (AC1)

**Draft for non-linear wind vane correction model**

Andreas Rott

August 2023

The deflection of the wind direction by the rotor largely results from two factors. Firstly, by the rotation of the rotor and a corresponding counter-rotation of the flow, and secondly by the thrust of the rotor if it is not directly aligned with the flow. For the model shown here, we assume that both causes are independent of each other and additive. This model is limited to the deflection caused by the thrust of the rotor.

The true wind direction is denoted by $\nu \in [-180°, 180°)$ and the wind direction measured by the wind vane by $\mu \in [-180°, 180°)$. Starting from the rotor centre as the coordinate origin, the wind vector in front of the rotor can be represented by the components $u_{\mathrm{ref}}$ (along the rotor axis, $x$-axis) and the lateral component $v_{\mathrm{ref}}$ (along the $y$-axis) (the $w$-component along the vertical upwards $z$-axis is neglected). It holds that

$$\tan(\nu) = \frac{v_{\mathrm{ref}}}{u_{\mathrm{ref}}}.$$

Behind the rotor, the wind direction consists of the $u_{\mathrm{vane}}$ and $v_{\mathrm{vane}}$ components. The $v$-component remains unchanged, $v_{\mathrm{vane}} = v_{\mathrm{ref}}$. The $u$-component is decelerated by the axial induction $a$ of the rotor: $u_{\mathrm{vane}} = u_{\mathrm{ref}} \cdot (1-a)$. Since the wind vane is not directly in the rotor plane we added another parameter $s$, which has to be optimized based on the data: $u_{\mathrm{vane}} = u_{\mathrm{ref}} \cdot (1 - sa)$ This results in the following for the measured wind direction deviation:

$$\tan(\mu) = \frac{v_{\mathrm{vane}}}{u_{\mathrm{vane}}} = \frac{v_{\mathrm{ref}}}{u_{\mathrm{ref}} \cdot (1 - sa)} = \frac{v_{\mathrm{ref}}}{u_{\mathrm{ref}}} \cdot \frac{1}{1 - sa} \Rightarrow \mu = \arctan\left(\tan(\nu) \cdot \frac{1}{1 - sa}\right)$$

we substitute $a$ with the thrust coefficient $c_{\mathrm{T}}$ with the relation $a = \frac{1}{2} - \frac{1}{2}\sqrt{1 - c_{\mathrm{T}}}$.

And the thrust coefficient gets a dependence on the misalignment $c_{\mathrm{T}} = c_{\mathrm{T0}} \cdot \cos(\nu)^p$ where $p$ is a parameter that theoretically should be close to 2, but can be optimized based on the given data and $c_{\mathrm{T0}}$ is the thrust coefficient for perfect alignment of the rotor into the wind. Thus, overall, the model is given by:

$$\mu_{s,p}(\nu) = \arctan\left(\tan(\nu) \cdot \frac{1}{1 + \frac{s}{2} \cdot \left(\sqrt{1 - c_{\mathrm{T0}} \cdot \cos(\nu)^p} - 1\right)}\right).$$

Figure 1 shows a comparison of the non-linear model (red) and the linear model (black). The parameters of the non-linear model are set to $s = 1, p = 2$, and the linear model has a slope of $c = 1.26$. It can be seen that for smaller misalignment the amplification of the wind direction deviation is stronger than the linear model and for larger misalignment the amplification gets weaker converging to the bisector. To answer the question whether the difference between the non-linear and the linear model within the meaningful range of $\nu \in [-30°, 30°]$ is large enough to justify the increased complexity of the model, further research is needed.

Unfortunately, this model can only be inverted numerically in the domain of definition.

[Figure]

Figure 1: Comparison of non linear model (red), linear model with slope of 1.26 (black) and the bisector (grey)

---

## Author Response (AR1)

We want to thank everyone for their constructive feedback.

In the following section, we address each of your comments and suggestions individually. Corresponding revisions of the manuscript will be referred to.
* * *
**Referee 1**

Thank you for the concise feedback. We have proofread our manuscript again and corrected typos to the best of our ability.
* * *
**Referee 2**

Thank you for the comments, which we will address in the following. Here we will paraphrase your feedback, preceded by RC2, and then give our answer, preceded by AC.

RC2-1:

use "correct" in L5 instead of "to account for"

AC-1:

done

RC2-2:

(L15) explain in more detail, why the alignment is a crucial parameter.

AC-2:

We have added the following sentence to emphasize the importance of the alignment of the rotor to the wind direction.

"Only if the turbine is oriented into the wind, it can reach its maximum possible power coefficient. Even with relatively small misalignment, its conversion of the kinetic energy in the wind into electricity is impaired. In addition, this imposes more uneven forces on the blades, which can result in increased fatigue loads. In certain situations, intentional misalignment of the turbine can be used to manipulate the wake so that downstream turbines are less affected. We refer to this approach as Active Wake Deflection and will discuss it briefly below. However, even for this particular mode of operation, it is therefore essential to accurately estimate the alignment of the wind turbine. The standard procedure to determine the alignment

involves using one or two wind vanes to detect deviations from the wind direction and adjust the yaw angle of the turbine through an active yaw manoeuvre accordingly."

RC2-3:

(L49-52) Phrase the questions in a more understandable way

AC-3:

1. Is there a systematic error in wind vane readings when utility-scale wind turbines are not aligned with the wind direction, and how can this error be described?

2. Can the inaccuracies in wind vane measurements be corrected using operational data, both with and without external reference measurements?

3. What impact does correcting the wind vane during yaw misalignment have on the overall performance of a wind turbine?

RC2-4:

(L104) Why was a static transfer function used and not a dynamic transfer function?

AC-4:

As far as we understand this question, a dynamic transfer function could consider the time dependence of the wind direction misalignment or the inclusion of additional parameters. For example, that the correction factor is not constant, but depends on wind direction change rate, its integrated mean, or other parameter. Examples comprise the wind speed, the atmospheric stability (TI, Monin-Obukhov length, wind shear, veer), the absolute wind direction in case of complex terrain, or the operation mode of the turbine (normal operation / curtailed operation).

Such parameters could potentially have an influence on the correction factor and are therefore worth additional investigations.

However, we chose to make the transfer function as straightforward as possible while still obtaining convincing results in the full-scale experiment, mainly for two reasons:

First, we believe that the most important parameter affecting the corrections factor is the thrust coefficient, which is relatively constant in the partial load range. We filtered for wind speeds and found that the estimated correction factor did not change significantly over the partial load range. For wind speeds greater than the rated wind speed and where the thrust coefficient decreases, the correction factor approaches a value of 1. Unfortunately, the amount of usable measurement data for

these situations was too small to obtain statistically valid results.

The second reason is that each new parameter adds another dimension of complexity and requires thorough investigation and sufficient data, in some cases, extra measurement devices, for example, in the case of atmospheric stability). To be able and allowed to implement the transfer function into the wind turbine controller, the function needed to be simple and convincing.

After our initial results have shown promising benefits for wind turbine operation, we hope that other manufacturers will feel encouraged to integrate similar or even more complex transfer functions into their turbines' control system and make the effort to test them.

RC2-5:

Rephrase L111

AC-5:

We reformulated the clause:

"As a result, wind vane data typically incorporates an adjustment to account for this deviation. Consequently, our subsequent analysis focusses on the correction factor $c$, with the offset factor $b$ being set to 0 degrees."

RC2-6:

What is the impact of uncertainty in the measurement of the met mast on the results?

AC-6:

The uncertainty in the measurement of the met mast means that we do not have a true reference for the wind direction. This complicates the analyses because we cannot simply estimate the uncertainty of the wind vane by comparing with a reference value. The cause of measurement errors in the correlation of both systems cannot clearly be identified between the two measurement systems and consists most likely of errors in both systems. As we explained in the paragraph about the ODR, standard statistical tools like ordinary linear regression lead to false results since these require having an undisturbed value (predictor, which is commonly plotted on the x-axis) and a response variable (on the y-axis, which can have uncertainty). In our case, both variables have uncertainties, so neither is the undisturbed predictor for the other. The undisturbed wind direction, which is unknown, is the predictor variable for both.

Fortunately, methods like ODR still allow us to identify correlations between the two

variables. And since we can assume that the uncertainty in the measurement of the met mast is independent of the yaw angle of the turbine, we can use them to characterize the influence of the yaw angle on the measurement of the wind vane. To strengthen this argument, we included the results of the CFD Simulation, where we know the true wind direction and see the same behaviour of a potential wind vane. Therefore, we conclude that using the met mast measurements is valid.

RC2-7 to 10:

Missing commas and typos

AC-7 to 10:

We corrected the corresponding passages.
* * *
**Community Comment 1**

Dear Johannes Schreiber,

Thank you very much for taking the time and effort to review the preprint and provide your extremely valuable and helpful comments. In the following, we will address your comments individually.

CC1-1:

Improve readability of L161

AC-1:

We have rewritten the sentence in the following way:

"In order to determine the influence of the rotor's thrust on the wind vane of the wind turbine, only situations in which the wind turbine was operated in the partial load range and without curtailment were considered for the comparison of the measured values."

CC1-2:

"The yaw control ensures that the yaw angle after the yaw manoeuvre corresponds to the measured wind direction before the yaw" Is this valid for every yaw controller? If a yaw controller yaws until the (filtered) relative wind direction becomes zero, the approach may not work as intended. I imagine that larger turbines, with low yaw speed implement such a control.

AC-2:

First, we would like to clarify how this is to be understood here: The filtered wind vane measurement before the yaw manoeuvre is used to trigger a yaw manoeuvre and determine the new target angle of the nacelle. The measurements during the yaw manoeuvre are NOT used; only after the yaw manoeuvre the measurements are recorded again and filtered for triggering the next yaw manoeuvre.

If the controller also uses the measurements during the yaw process in the filter, then the system would overshoot to set the filtered value to zero. (We assume this is meant in the second part of the question, so we agree with that). In control engineering terms, the yaw controller would thus be a pure I-controller (Integral, without proportional (P) or derivative (D) parts). It would, of course, be possible to create a complete PID controller that theoretically would not overshoot, but it has proven best only to use strongly filtered values of the wind vane since the individual measurements fluctuate very strongly, which would make it very difficult to balance the P and D portion in the controller.

For precisely this reason, the target value is determined before the yawing process, and the measurements are discarded during yawing.

It is challenging to say whether all yaw controllers work on precisely this principle, as wind turbine manufacturers commonly do not share yaw control algorithms. To the best of our knowledge, this is how it works in all the turbines we have dealt with. This reactive behaviour of the yaw control thus implicitly assumes a persistence model for the wind direction, which is a logical assumption given the lack of better alternatives. Therefore, the research on proactive yaw controllers is fascinating and can utilize information gathered upstream. (See Sengers, B. A. M., Rott, A., Simley, E., Sinner, M., Steinfeld, G., and Kuehn, M.: Increased power gains from wake steering control using preview wind direction information, Wind Energ. Sci. Discuss. [preprint], https://doi.org/10.5194/wes-2023-59, in review, 2023.)

CC1-3:

Maybe include binned mean values to show a possible non-linear dependency between the wind direction deviation reference and wind direction deviation measured by the wind vane.

AC-3:

We agree that investigating non-linearity is exciting research, which can be

necessary for active wake deflection.

In the context of this paper, we decided against it because the focus here is on standard operation, and the linear modelling of the relationship between wind direction deviation and the wind vane measurement seems sufficient for this operation. We did not want to complicate the paper or make it even longer unnecessarily, and more data with intentional misalignment should be investigated for an investigation with due diligence.

However, we like to use this discussion to present a few more thoughts/suggestions that we have made on this topic:

Aggregating the data in Fig. 7 and Fig. 8 by binning and thus investigating the correlation sounds logical and was one of our first approaches. However, this leads to the same problem as with linear regression. The mapping of the wind direction deviation measured by the wind vane to the x-axis and the wind direction deviation measured by the met mast to the y-axis is arbitrary and can be swapped freely since both measurements have uncertainties, and thus neither is a valid predictor of the other.

As with linear regression, binning implicitly assumes no measurement uncertainty on the binned value. If we apply binning to the x-axis, we get a different result than if we swap the axes or use binning to the y-axis. Therefore, the validity of the binned mean values is questionable.

A reasonable method for investigating the linear or non-linear relationship would be the principal component analysis or the Proper Orthogonal Decomposition. Here, the data are projected onto a vector space that maps the variances sorted by size. The number of dimensions of the new vector space does not change, so two dimensions remain. The first dimension (principal component/mode) represents the largest variances within the scatterplot. This forms a straight line through the scatterplot, minimising the squared orthogonal distances to the scatter points.

The ODR line, which we have determined, fulfils exactly this property and is, therefore, the first principal component / the first mode. This method shows, first, only a linear relation and, in fact, the strongest one. To analyse a possible non-linear relationship, the second dimension of the new vector space can be examined, i.e., the residual or error that remains in the ODR. The total error will yield a distribution with a mean of 0 (this is the case by definition). But in the case of a pure linear correlation, the error should be randomly distributed over the entire measurable range of the ODR line and have no discernible structure. In the case of a non-linear correlation, this should be recognisable if we consider, for example, now binned mean values of the residual over the ODR straight line.

For modelling the non-linear relationship, we have come up with a model that makes sense, but more data with larger misalignments will be needed for a thorough investigation that proves the advantages of this model over the linear model. We have briefly outlined the model in the appendix of this discussion [Draft of non-linear wind vane correction model].

CC1-4:

Uncertainties to the yaw actuation and distances?

AC-4:

The number of average actuations is simply calculated by dividing the total number of actuations within the period by the duration, extrapolated to 10 min. Different methods could be used to get an idea of the distribution, but their results vary slightly. For example, divide the time series into 10 min sections and look at the number of yaw manoeuvres in each interval. The average value of these data also corresponds to the average value given here. The fluctuation is notable here for the interpretation since this data basis permits only whole numbers. It would make more sense to look at the distribution of the number of yaw manoeuvres in 10 min for each day or, for example, for all four hours. This might show further correlations between, e.g., atmospheric stability, turbulence intensity, and the number of yaw manoeuvres.

The bootstrapping method is beneficial for calculating the standard error of the mean when only small amounts of data are available.

CC1-5:

Is yaw distance per 10 min equivalent to the more commonly used actuator duty cycle (ADC)

AC-5:

The Actuator Duty Cycle (ADC) and the Yaw Distance per 10 min specified here are very. Due to the inertia of the nacelle the rotational speed of the nacelle is not constant during acceleration and deceleration, and therefore the yaw distance does not correlate 100 % with the duration of the yaw manoeuvre and thus not with the ADC. (Small yaw manoeuvres take more time per degree of turn, relatively speaking.)

But this is a useful hint, which has been added to the script as the following paragraph in Chapter 3.3.1:

"The yaw distance, in general, is closely related to the actuator duty cycle (ADC) of the yaw controller. The BARD5.0 turbine yaws at an average rotational speed of about $0.75^\circ$ per second, so the average yaw distance per 10 min of $13.66^\circ$ for normal operation takes approximately 18.21 seconds, which means that the yaw motor is active about $3\,\%$ of the time. Due to the inertia of the nacelle, the rotation speed is not entirely constant; therefore, this conversion is only an approximation. Thus, in this evaluation, we only use the average yaw distance per 10 min and not additionally the ADC."